# A Character-Level Length-Control Algorithm for Non-Autoregressive Sentence Summarization

**Puyuan Liu, Xiang Zhang, Lili Mou**[*]
Dept. Computing Science, Alberta Machine Intelligence Institute (Amii)
University of Alberta, Canada
[*]Canada CIFAR AI Chair, Amii
{puyuan, xzhang23}@ualberta.ca, doublepower.mou@gmail.com

## Abstract

Sentence summarization aims at compressing a long sentence into a short one that keeps the main gist, and has extensive real-world applications such as headline generation. In previous work, researchers have developed various approaches to improve the ROUGE score, which is the main evaluation metric for summarization, whereas controlling the summary length has not drawn much attention. In our work, we address a new problem of explicit character-level length control for summarization, and propose a dynamic programming algorithm based on the Connectionist Temporal Classification (CTC) model. Results show that our approach not only achieves higher ROUGE scores but also yields more complete sentences.[1]

## 1 Introduction

Sentence summarization aims at compressing a long sentence into a short one that conveys the main idea of the input. It is an established setting of summarization, and has extensive real-world applications such as generating news headlines [26, 30, 8] and being a key component of document summarization [40, 21]. Previous research has focused on improving the ROUGE score, a widely used evaluation metric for summarization [17], while less attention has been paid to controlling the length of the summary.

Recently, researchers argue that length control is the key to summarization, since it is typically required by real-world applications [20]. Moreover, the ROUGE score is found to be sensitive to the summary length [33, 32], and summarization systems can achieve higher scores by simply generating longer output. Previous work mainly addresses length control in the word level, i.e., restricting the summary length by a pre-defined number of words [33, 18].

In this paper, we address explicit length control in the character level for summarization, which is a different setting from previous work. In other words, we restrict the summary length by the number of characters, such as letters, punctuation marks, and whitespaces. We observe that this is a more realistic setting in real-world applications than word-level length control. For example, the headline shown in a mobile app or web page is constrained by the screen width (roughly speaking, the number of characters), rather than the number of words. Likewise, a commercial LED display allows a certain number of characters; ideally, the text shown should fit the character-level budget, or otherwise it will be scrolling, making it difficult to read. However, the character-level constraint is unlikely to be satisfied if we only perform word-level control, despite the positive correlation between word and character lengths. Moreover, character-level length control is a common evaluation setting for summarization systems. For example, a sub-task in the DUC evaluation requires the maximum target length to be 75 bytes [27]. Such an important setting, unfortunately, is not adequately addressed

---

[1]Our code, model, and output are released at: https://github.com/MANGA-UOFA/NACC

36th Conference on Neural Information Processing Systems (NeurIPS 2022).

in previous summarization studies. Our work largely bridges the gap between the methodological summarization research and its evaluation.

We further observe that controlling the summary length by characters cannot be easily addressed by previous approaches. For example, truncating is able to explicitly control the length, but the resulting summary is incomplete; Takase and Okazaki [36] feed length embeddings into the model as input, but such an approach cannot control the summary length in an explicit manner; and Schumann et al. [33] perform constrained discrete optimization by selecting a certain number of words from the source text as the output, but their generated summaries may vary to a large extent in terms of the number of characters.

To this end, we propose NACC, a Non-Autoregressive summarization model with Character-level length Control. We adopt a non-autoregressive approach because it generates all tokens in parallel and is much faster than autoregressive models. More importantly, we observe that non-autoregressive models predict the output words independently. Such predicted probabilities are thus local, which provides us with the unique opportunity to design a dynamic programming algorithm to constrain the summary length. Specifically, we formulate length control as a knapsack-like problem, where the weight is the number of characters in a token and the value is the predicted probability of the token. In this way, we are able to explicitly control the summary length at the character level, while retaining the completeness of the output text.

We evaluated our NACC model on the Gigaword headline generation [10] and DUC2004 [27] datasets in two settings: supervised and unsupervised. In the latter setting, NACC learns from the pseudo-reference given by an unsupervised word-extraction method based on discrete search [33]. Experiments show that NACC establishes the state-of-the-art performance of non-autoregressive summarization under various target lengths in both settings; NACC even outperforms autoregressive Transformers [37] in the unsupervised setting, where the input and output have stronger correspondence. These all confirm the effectiveness of our length-control algorithm. Regarding inference efficiency, we show that non-autoregressive models without length control are 10 times faster than autoregressive ones; even with our length-control dynamic programming, NACC is still several times more efficient. Further, our NACC is capable of length-transfer generation, i.e., generating summaries of different lengths from the training targets.

## 2 Related Work

**Non-Autoregressive Generation.** Non-autoregressive (NAR) models [12] predict target tokens in parallel and thus enjoy a much higher inference efficiency than autoregressive (AR) ones, which generate output tokens sequentially. NAR generation is more difficult than AR, because there lacks dependencies among the outputs. Previous work addresses the dependency issue by iterative refinement [16, 22, 3, 13] or structured decoding with the conditional random field (CRF) [35, 34, 4].

The Connectionist Temporal Classification (CTC) algorithm [11] addresses a common problem in NAR generation, namely, token repetition, by merging consecutive identical tokens (unless separated by an empty token). We follow our previous work [18] and adopt CTC for NAR summarization, allowing empty tokens scattering over the entire sentence to generate a short summary.

NAR models are traditionally thought to generate worse-quality text than AR models. However, we have the unique insight that NAR also brings new opportunities for designing length-control algorithms: since the model predicts output independently, the decoding problem can be divided into shared sub-problems for dynamic programming.

**Text Summarization.** Summarization systems can be generally categorized into two types: extractive and abstractive. Extractive methods output a summary by extracting important sentences or clauses from the source text [5, 23, 15, 44], while abstractive methods generate summaries with new expressions [25, 28, 9].

Depending on the availability of training data, we may categorize summarization systems into supervised and unsupervised ones. Supervised methods typically follow a sequence-to-sequence (seq2seq) paradigm [43, 19]. Recently, unsupervised summarization is drawing increasing attention because it does not require parallel training data. Yang et al. [42] use the lead approach (selecting the first several words or sentences) as pseudo-groundtruth for seq2seq training. Alternatively, cycle consistency is adopted as the training objective for unsupervised summarizaiton [38, 2]. Schumann et

al. [33] generate a summary by maximizing a heuristically defined scoring function (involving fluency and semantic similarity) with word-level extraction. We consider both supervised and unsupervised settings in our work.

Controlling the output length is crucial for deploying summarization systems to real-world applications. In early extractive summarization research, truncating is adopted for fair comparison [23, 24, 15], but the resulting summary may not be complete sentences. As pointed out by [32, 33], however, the length-control problem is not adequately addressed for abstractive summarization in the neural era, probably because researchers do not want to hurt the completeness. For example, length information [20, 36] and soft penalty [34] are adopted to encourage short summaries, but they cannot control the length in an explicit way. Very recently, Schumann et al. [33] address the length-control problem by extracting a certain number of source tokens as the summary. Our previous work [18] designs a CTC-based algorithm that controls the number of output words explicitly. However, these approaches cannot be applied to the character level. Our paper proposes a character-level length-control algorithm that can explicitly constrain the number of characters within a given budget, which is different from all previous work.

## 3 Approach

In this section, we first introduce the CTC-trained NAR model for summarization (§3.1). Then, we propose a dynamic programming algorithm to explicitly control the number of characters (§3.2).

### 3.1 A Non-Autoregressive Model for Summarization

In the summarization task, the goal is to compress a source text $\mathbf{x} = (x_1, x_2, \ldots, x_S)$ into a shorter text $\mathbf{y} = (y_1, y_2, \ldots, y_T)$, while preserving the key information.

**Encoder-Only Architecture.** Our work follows recent non-autoregressive summarization methods [34, 18], utilizing the encoder-only Transformer [37] as the base model (Figure 1a). It is essentially a stack of Transformer layers, the last of which predicts all words simultaneously. Thus, it has a much higher inference efficiency than autoregressive models.

Formally, let $E^{(i)} \in \mathbb{R}^{S \times d}$ be the contextual representation of the input $\mathbf{x}$ at the $i$th Transformer layer, where $S$ is the number of tokens in $\mathbf{x}$ and $d$ is the dimension. Specially, $E^{(0)}$ is the word and positional embeddings of input tokens. Each layer is a Transformer block [37] based on its predecessor, which can be denoted by $E^{(i)} = \text{Layer}^{(i)}(E^{(i-1)})$, for $i \geq 1$.

At the last layer $E^{(L)}$, a $\text{softmax}$ function is applied to predict the probability at every prediction slot independently, given by $P_s(\cdot|\mathbf{x}) = \text{softmax}(W \boldsymbol{e}_s^{(L)})$ for slots $s = 1, \cdots, S$, where $\boldsymbol{e}_s^{(L)}$ is a column vector transposing the $s$th row of matrix $E^{(L)}$, and $W$ is a weight matrix.

Such an encoder-only architecture is able to capture the strong correspondence between the input and output in the summarization task and outperforms encoder–decoder non-autoregressive models, shown by our previous work [18].

**CTC Training.** Training the above encoder-only architecture requires padding the target with empty tokens $\epsilon$; otherwise, the output will have the same length as the input, and cannot be a summary. Gu et al. [12] pad $\epsilon$s at the end of the target, but this hinders the input–output correspondence.

Instead, we follow our previous work [18] and train our model with the Connectionist Temporal Classification (CTC) [11], allowing $\epsilon$s to scatter over the entire target summary as appropriate. During inference, the CTC approach removes $\epsilon$s to generate a shorter text as the summary. In addition, CTC merges consecutive identical tokens that are not separated by $\epsilon$, because non-autoregressive generation suffers from the word-repetition problem [12, 31]. We denote the CTC operation as $\Gamma$, for example, $\Gamma(aa\epsilon\epsilon abb\epsilon) = aab$.

For training, CTC maximizes the marginal likelihood of token sequences (including $\epsilon$) that can be reduced to the target text $\mathbf{y}$:

$$P(\mathbf{y}|\mathbf{x}) = \sum_{\mathbf{w}:\Gamma(\mathbf{w})=\mathbf{y}} P(\mathbf{w}|\mathbf{x}) = \sum_{\mathbf{w}:\Gamma(\mathbf{w})=\mathbf{y}} \prod_{s=1}^{S} P_s(w_s|\mathbf{x}) \tag{1}$$

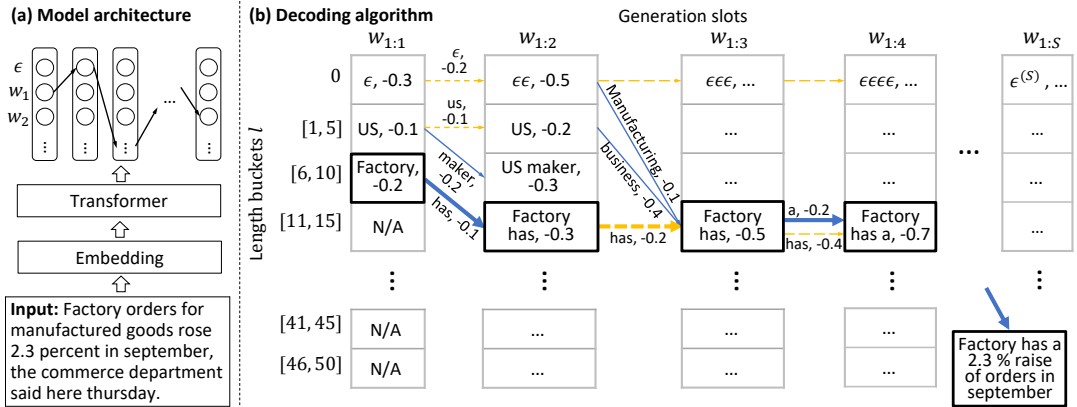

Figure 1: Overview of our NACC approach. Dashed yellow arrows refer to transitions that do not increase the summary length, while solid blue arrows refer to the increase of length. Thick arrows and blocks refer to the selected path by CTC. Due to the space limit, $\epsilon$ is omitted in the predicted sentence, and we use $\epsilon^{(S)}$ to denote a sequence of $S$-many $\epsilon$s. The number demonstrates the value (i.e., log-probability) of a word.

where $P(\mathbf{w}|\mathbf{x})$ is the probability of generating a token sequence $\mathbf{w} = w_1, \cdots, w_S$, and $P_s(w_s|\mathbf{x})$ is given by the non-autoregressive model. Although a brute force summation in Eqn. (1) is intractable, it can be efficiently computed by dynamic programming. We refer interested readers to Alex et al. [11] for the details of the above marginalization in CTC training.

## 3.2 The Proposed Algorithm for Character-Level Length-Control Decoding

One of our main contributions is to propose a dynamic programming (DP) algorithm for character-level length control. As mentioned in §1, controlling the number of characters in the output is important for summarization, since real-world applications usually involve character-level length constraints.

Our insight is that a non-autoregressive model predicts probabilities independently, so the length-control problem can be divided into shared sub-problems, making DP algorithms feasible. We formulate character-level length control as a knapsack-like problem. We treat the number of characters in a word (plus one) as the weight,[2] denoted by $u(w)$ for the word w, and the predicted log-probability as the value $v_s(w) = \log P_s(w|\mathbf{x})$ for the prediction slot $s$. Our goal of character-level length-control summarization can be formulated as

$$\underset{w_1, \cdots, w_S}{\text{maximize}} \sum_{s=1}^{S} v_s(w_s), \qquad \text{subject to} \sum_{\substack{\mathbf{y} = \Gamma(w_1, \cdots, w_S)}}^{y \in \mathbf{y}} u(y) < U \qquad (2)$$

where $U$ is the total length budget. Here, the value is the sum of the log-probability of every generation slot including $\epsilon$, whereas the length is said in terms of the words of the CTC-reduced sequence $\mathbf{y} = \Gamma(w_1, \cdots, w_S)$.

We observe that handling every possible integer weight (i.e., length) as in a standard knapsack algorithm may slow down the inference. Thus, we divide the lengths into buckets for efficient inference. Formally, let the $l$th bucket cover the length ranging from $\alpha \cdot (l-1) + 1$ to $\alpha \cdot l$ characters, where $\alpha$ is a hyperparameter controlling the bucket size. We denote by $\mathbf{d}^{s,l} = d_1^{s,l} \cdots d_s^{s,l}$ the most probable $s$-token sequence that is reduced to a summary in the $l$th length bucket. Specially, we let $\mathbf{d}^{s,0}$ mean that the reduced summary has zero words.

The initialization of $\mathbf{d}^{s,l}$ fills in the DP table for $l = 0$ and $s = 1$.

---

[2]For the purposes of research, we assume every word is appended with another character, namely, a whitespace. In real applications, our algorithm can be applied to any measure of length, such as the display width of a word in some font.

- For $l = 0$, we must have $\mathbf{d}^{s,0} = \epsilon \cdots \epsilon$ ($s$-many), because $l = 0$ means no non-$\epsilon$ word has been generated. (First row of Figure 1b.)
- For $s = 1$, we have

$$\mathbf{d}^{1,l} = \begin{cases} \epsilon, & \text{if } l = 0 \\ \underset{\mathrm{w}:u(\mathrm{w})\in[\alpha\cdot(l-1)+1,\alpha\cdot l]}{\mathrm{argmax}} v_1(\mathrm{w}), & \text{if } l > 0 \end{cases} \tag{3}$$

Here, $l = 0$ is the same as the previous bullet item. For $l > 0$, we select the most probable word for each length bucket according to the value $v_1(\cdot)$, i.e., the predicted log-probability of the first generation slot. (First column of Figure 1b.)

The DP recursion is to compute $\mathbf{d}^{s,l}$ based on a newly predicted token $\mathrm{w}_s$, assuming its top-left sub-table is filled. This involves three scenarios:

- Case 1: $\mathrm{w}_s = \epsilon$. In this case, the new word is $\epsilon$. Thus, the index for generation slots increases from $s - 1$ to $s$, but the summary length does not change. We denote $\mathscr{D}_1^{s,l}$ as the set containing the candidate sequence, given by

$$\mathscr{D}_1^{s,l} = \left\{ \mathbf{d}^{s-1,l} \oplus \epsilon \right\} \tag{4}$$

where $\oplus$ denotes string concatenation. (See yellow dash arrows in Figure 1b.)
- Case 2: $\mathrm{w}_s \neq \epsilon$, but $\mathrm{w}_s = \mathrm{d}_{s-1}^{s-1,l}$. In other words, the candidate non-$\epsilon$ word $\mathrm{w}_s$ for the $s$th slot is identical to the last token of $\mathbf{d}^{s-1,l}$. Since repeated tokens are merged during CTC decoding, the output length index $l$ is unchanged. We form this sequence as a set:

$$\mathscr{D}_2^{s,l} = \left\{ \mathbf{d}^{s-1,l} \oplus \mathrm{d}_{s-1}^{s-1,l} \right\} \tag{5}$$

(Also see yellow dash arrows in Figure 1b.)
- Case 3: $\mathrm{w}_s \neq \epsilon$ and $\mathrm{w}_s \neq \mathrm{d}_{s-1}^{s-1,l'}$ for some $l' \leq l$. That is, $\mathrm{w}_s$ is neither $\epsilon$ nor repetition, and thus the summary length will be increased from bucket $l'$ to $l$. We denote this candidate set by

$$\mathscr{D}_3^{s,l} = \left\{ \mathbf{d}^{s-1,l'} \oplus \mathrm{w}_s \ : \ \left( u(\mathrm{w}_s) + \sum_{\mathrm{d}\in\mathbf{d}^{s-1,l'}} u(\mathrm{d}) \right) \in [\alpha \cdot (l-1) + 1, \alpha \cdot l], \right. \tag{6}$$

$$\left. \mathrm{w}_s \neq \epsilon, \mathrm{w}_s \neq \mathrm{d}_{s-1}^{s-1,l'}, \text{ and } l' \leq l \right\}$$

(See blue arrows in Figure 1b.)

Then, our DP finds the most probable sequence at each recursion step:

$$\mathbf{d}^{s,l} = \underset{\mathbf{d}\in\mathscr{D}_1^{s,l}\cup\mathscr{D}_2^{s,l}\cup\mathscr{D}_3^{s,l}}{\mathrm{argmax}} \sum_{s=1}^{S} v_s(\mathrm{d}_s) \tag{7}$$

where $\mathrm{d}_s$ is the $s$th token of a sequence $\mathbf{d}$ from the three candidate sets above.

**Theorem 1.** *(1) If the bucket size $\alpha = 1$ and consecutive repetitions are not merged, then $\mathbf{d}^{S,T}$ is the most probable sentence of $T$ characters given by the $S$ prediction slots. (2) If $\alpha \neq 1$ or repeating tokens are merged, our algorithm may not be exact. (See Appendix A for the proof.)*

**Discussion.** Our DP algorithm is inspired by the standard 0-1 knapsack problem [1], but also differs in several significant ways. First, we merge consecutive tokens during CTC decoding; this establishes some dependencies among different generation slots, and thus exact inference with DP is not possible. Second, our value function is non-stationary, as it changes over time. We require that every slot should select a token, either $\epsilon$ or a word. In both cases, the token's value is added to the total value. Therefore, our algorithm is compatible with negative values, namely, log-probabilities in our application, because only the relative difference matters for the value function.

## 4 Experiments

### 4.1 Setup

**Datasets.** Our model is evaluated on the Gigaword headline generation [30] and DUC2004 datasets [27]. The Gigaword dataset comprises pairs of news articles and titles; we follow the

standard setting [30, 33, 18] and adopt the first sentence of each news article as the input and consider the news title as the summary. In total, the dataset contains 3.8M, 198K, and 1951 samples for training, validation, and test, respectively. On the other hand, the DUC2004 dataset has 500 paired samples and is designed for test only; its performance is obtained by models trained on Gigaword.

**Metrics.** We follow previous work [33, 30, 34] and evaluate the output by ROUGE scores [17]: ROUGE-$n$ evaluates $n$-gram overalpping, whereas ROUGE-L evaluates the longest common sequence. We follow the convention and adopt ROUGE F1 for the Gigaword dataset [38, 33, 18] and ROUGE Recall for the DUC2004 dataset [6, 39].

**Implementation Details.** We use a Transformer encoder as the base model, which has 6 layers and 8 attention heads for each layer, following the settings in [37]. The dimensions are 512 and 2048 for the attention and feed-forward modules, respectively. Each training batch contains samples amounting to 4K tokens. The learning rate is chosen from {1e-4, 5e-4} by validation, and we ran 100K gradient updates for the unsupervised setting, but 400K updates for the supervised setting. Note that, in the unsupervised setting, our model learns from an extractive search-based method [33], and thus its training is relatively easy. By contrast, the supervised setting takes human-written sentences as the training targets, which are more abstractive and difficult to train with. All experiments were run on an i9-9940X CPU and an RTX6000 GPU.

For our length-control algorithm, we adopt a bucket size of 4, and only consider the most probable 20 words for every generation slot (cf. $w_s$ in Eqn. 6) due to efficiency concerns.

## 4.2 Results and Analyses

**Results on Gigaword Headline Generation.** Table 1 presents the performance on the Gigaword test set in both supervised and unsupervised settings. For the supervised setting, we train with the groundtruth references, which contain 51.29 characters on average. In the unsupervised setting, the training targets of machine learning models (Rows 9–13) are the output summaries given by an unsupervised search-based method [33]; specifically, we adopt the standard 8-word setting, which counts to 48.75 characters on average. Based on these, we set our length constraint to 50 characters. We observe that our proposed algorithm is the only machine learning-based approach that can perform explicit character-level length control. For fair comparison, we truncate the output of other methods to satisfy the length constraint.

As seen, NACC even with truncating outperforms all other non-autoregressive (NAR) models [34, 29, 41] in both settings. Specifically, Su et al. [34] emit multiple end-of-sequence tokens at the end of the output sequence to generate a shorter summary than the source text (Rows 1 & 9); Qi et al. [29] propose to pretrain a summarization system in an autoregressive manner, and gradually adapt it to the NAR setting (Rows 5 & 10); and Yang et al. [41][3] propose a two-step strategy of autoregressive part-of-speech (POS) prediction and non-autoregressive summary generation (Rows 6 & 11). Our NACC trained by CTC is able to surpass all these methods, demonstrating the effectiveness of CTC training for summarization. Besides, NACC outperforms its search-based teacher model [33], showing that learning can smooth out the search noise.

Equipped with the length-control algorithm, our NACC model has a significant boost in terms of ROUGE scores. In a fair comparison with the same base architecture, the length-control algorithm alone improves NACC (truncate) by ~3 points. Our full model achieves an improvement of more than 4 total ROUGE points compared with previous state-of-the-art NAR methods.

Regarding the inference efficiency, NACC with truncating is faster than previous NAR models. Even with the length-control algorithm, our NACC is still in the same ballpark, being ~500x faster than the search-based method.

Additionally, we design a variant of the unsupervised summarization method based on [33], where we directly impose the character-level length constraint during each search step (Row 8). We find this approach outperforms truncating word-constrained search (Row 7), but is much worse than our machine learning-based NACC with the length-control algorithm.

---

[3]Yang et al. [41] only provided execution commands in their GitHub repo, but no training code. We emailed the authors, but have not obtained the code either. The reported results are based on our best replication.

| Setting | # | | Approach | Len | ROUGE F1 | | | | Time |
|---|---|---|---|---|---|---|---|---|---|
| | | | | | R-1 | R-2 | R-L | ΔR | |
| Supervised | 1 | | Su et al. [34] (truncate) | 38.43 | 32.28 | **14.21** | 30.56 | 0 | 0.016 |
| | 2 | | Qi et al. [29] (truncate) | 27.98 | 31.69 | 12.52 | 30.05 | -2.79 | 0.019 |
| | 3 | NAR | Yang et al. [41] (truncate) | 35.37 | 28.85 | 6.45 | 27.00 | -14.75 | – |
| | 4 | | NACC (truncate) | 34.15 | 33.12 | 13.93 | 31.34 | 1.34 | **0.011** |
| | 5 | | NACC (length control) | 34.40 | **33.66** | 13.73 | **31.79** | **4.74** | 0.017 |
| Unsupervised | 6 | Baseline | Lead-50 chars | 49.03 | 20.66 | 7.08 | 19.30 | -9.23 | – |
| | 7 | Search | Schumann et al. [33] (truncate) | 45.45 | 24.98 | 9.08 | 23.18 | 0.97 | 9.573 |
| | 8 | | Char-constrained search | 44.05 | 25.30 | **9.25** | 23.43 | 1.71 | 17.324 |
| | 9 | | Su et al. [34] (truncate) | 45.24 | 24.65 | 8.64 | 22.98 | 0 | 0.017 |
| | 10 | | Qi et al. [29] (truncate) | 44.54 | 24.31 | 7.66 | 22.48 | -1.82 | 0.019 |
| | 11 | NAR | Yang et al. [41] (truncate) | 49.37 | 21.70 | 4.60 | 20.13 | -9.84 | – |
| | 12 | | NACC (truncate) | 47.77 | 25.79 | 8.94 | 23.75 | 2,21 | **0.012** |
| | 13 | | NACC (length control) | 47.03 | **27.45** | 8.87 | **25.14** | **5.19** | 0.025 |

Table 1: Performance on the Gigaword headline generation test set, where NAR stands for non-autoregressive. **Len:** Average number of characters in the predicted summaries. **R-1, R-2, R-L:** ROUGE-1, ROUGE-2, ROUGE-L. **ΔR:** The difference of total ROUGE (sum of R-1, R-2, and R-L) in comparison with the (previous) state-of-the-art NAR summarization system [34]. **Time:** Average inference time in seconds for one sample on an i9-9940X CPU and an RTX6000 GPU.

| # | | Approach | ROUGE Recall | | | | Time |
|---|---|---|---|---|---|---|---|
| | | | R-1 | R-2 | R-L | ΔR | |
| 1 | Baseline | Lead-75 chars | 22.52 | 6.50 | 19.74 | -4.97 | – |
| 2 | Search | Schumann et al. [33] (truncate) | 26.09 | **8.03** | 22.86 | 3.25 | 30.362 |
| 3 | | Char-constrained search | 26.30 | 7.95 | 22.78 | 3.30 | 31.540 |
| 4 | | Su et al. [34] (truncate) | 24.67 | 7.25 | 21.81 | 0 | 0.017 |
| 5 | NAR | Qi et al. [29] (truncate) | 22.79 | 5.91 | 20.05 | -4.98 | 0.018 |
| 6 | | NACC (truncate) | 26.43 | 7.86 | 22.66 | 3.22 | 0.012 |
| 7 | | NACC (length control) | **28.37** | 7.74 | **24.30** | **6.68** | 0.030 |

Table 2: Results on DUC 2004 dataset.

**Results on DUC2004.** Table 2 shows the performance of NACC on DUC2004, where we constrain the summary length to be at most 75 characters, following previous work [39, 2, 33]. Since the Gigaword training summaries have very different lengths from DUC2004 summaries, it is not straightforward to evaluate NACC in the supervised setting, as this involves length transfer. Therefore, we consider the unsupervised setting for DUC2004, where we use 13-word summaries (~80 characters) from [33] as our training targets. (We will have length-transfer experiments in later analysis.)

Experiments show that NACC with length control again largely outperforms previous NAR models, while retaining high inference efficiency. Results are consistent with the Gigaword experiment.

**Human Evaluation.** We additionally conducted human evaluations to investigate the effectiveness of our approach. Due to the limit of time and resources, we mainly considered the overall quality and completeness/fluency, as they are the key of our work but may not be adequately captured by automatic metrics such as ROUGE scores. For controlled comparison, we consider NACC with truncating and NACC with the proposed length-control algorithm.

We invited three human annotators to compare the output summaries on each of the 150 randomly selected samples from the Gigaword test set. We adopted the unsupervised setting in Table 1, where models were trained with 8-word summaries generated by [33]. The two systems' outputs for a

| | Decoding | Wins | Ties | Loses | $p$-value |
|---|---|---|---|---|---|
| Overall quality | Truncate | 18% | 44% | 38% | 0.0001 |
| | Length control | **38%** | 44% | **18%** | |
| Completeness & fluency | Truncate | 22% | 36% | 42% | 0.0002 |
| | Length control | **42%** | 36% | **22%** | |

Table 3: Human evaluation comparing truncating and length-control decoding of our NACC approach on 150 samples selected from the Gigaword headline generation dataset in the unsupervised setting. The $p$-value is given by a two-sided binomial test.

| Setting | # | Approach | | Len | ROUGE F1 | | | Time |
|---|---|---|---|---|---|---|---|---|
| | | | | | R-1 | R-2 | R-L | |
| Supervised | 1 | AR | Transformer (truncate) | 40.89 | **35.12** | **16.61** | **32.55** | 0.093 |
| | 2 | | Transformer (length control) | 39.73 | 34.10 | 15.65 | 31.64 | 0.095 |
| | 3 | NAR | NACC (truncate) | 34.15 | 33.12 | 13.93 | 31.34 | **0.011** |
| | 4 | | NACC (length control) | 34.40 | 33.66 | 13.73 | 31.79 | 0.017 |
| Unsupervised | 5 | AR | Transformer (truncate) | 46.62 | 26.31 | **9.33** | 24.29 | 0.092 |
| | 6 | | Transformer (length control) | 45.23 | 25.33 | 9.03 | 23.44 | 0.095 |
| | 7 | NAR | NACC (truncate) | 47.77 | 25.79 | 8.94 | 23.75 | **0.012** |
| | 8 | | NACC (length control) | 47.03 | **27.45** | 8.87 | **25.14** | 0.025 |

Table 4: Comparing autoregressive (AR) and non-autoregressive (NAR) models on the Gigaword headline generation test set. Our length-control algorithm requires the predicted probabilities to be independent, and thus is not compatible with AR models.

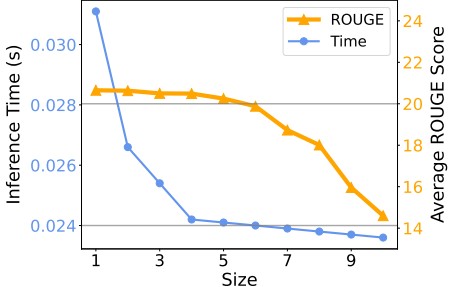

Figure 2: Performance of NACC with different bucket sizes.

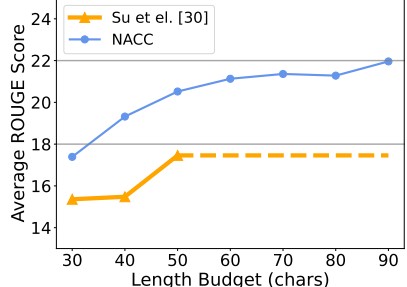

Figure 3: Length-transfer performance of NACC and Su et al. [34].

sample were blindly given to annotators in a random order; annotators then voted for the better summary or a tie, regarding the overall quality and completeness/fluency separately. We counted the votes for each decoding method on the two evaluation criteria, and show the percentage of wins/ties/loses in Table 3.

As seen, our length-control decoding has a dominant advantage over the truncating method in terms of both the overall quality and completeness/fluency. Further, this result is statistically significant based on a two-sided binomial test (ignoring ties), verifying that our length-control algorithm indeed improves the quality and completeness of the predicted summaries.

**Comparison with Autoregressive Models.** We are curious about how our non-autoregressive NACC is compared with autoregressive (AR) methods. Thus, we train a standard AR Transformer with truncating and length-control decodings, and show results in Table 4.

As seen, our length-control algorithm is not compatible with the AR Transformer and hurts the ROUGE scores (Rows 2 & 6). This is because our algorithm is based on dynamic programming and requires model outputs to be local, so that the length-control problem can be divided into shared sub-problems; however, the predicted probabilities of the AR Transformer depend on partial generation at previous time steps. Note that this is not a disadvantage of our approach, but shows that NAR generation provides unique opportunities for length control.

Admittedly, NACC has slightly lower ROUGE scores than AR Transformers in the supervised setting (Rows 1–4), because human-written summaries are flexible, causing difficulties for NAR training. Nevertheless, our NACC achieves much higher inference efficiency, and outperforms all previous NAR systems (recall Table 1).

In the unsupervised setting, our approach achieves higher performance than the AR Transformer, which is a very strong result. This is because we learn from a search-based method that extracts source words as the summary [33], and our CTC training—with blank tokens scattering over the whole output sentence as appropriate—can capture such strong correspondence between input and output. Moreover, the proposed length-control algorithm is able to maintain the summary completeness given the length constraint, achieving better ROUGE scores than NACC with truncating.

| |
|---|
| **Input:** singapore airline and delta air lines announced two differing strategies to upgrade their long-haul in-flight service for business travelers . |
| **Reference:** business travel : competing strategies ; crowded skies |
| **Supervised Setting:** |
| **AR Transformer (no control):** delta airlines upgrade service for business travel (50 characters) |
| **NACC (no control):**                delta to upgrade long-haul in-flight (36 characters) |
| **NACC (length control):**         delta air to upgrade long-haul in-flight service (48 characters) |
| **Unsupervised Setting:** |
| **AR Transformer (no control):** delta air lines differing strategies to upgrade their in-flight service for business travelers |
| **NACC (no control):**                delta air lines differing strategies to upgrade long-haul in-flight service for business travelers |
| **NACC (length control):**         delta air lines upgrade service business travelers |

Table 5: Example summaries for Gigaword headline generation, where gray words are truncated for fair comparison.

**Analysis of the Length Bucket.** Our dynamic programming is an approximate algorithm with a $\alpha$-sized length bucket (see Figure 1b and §3.2). Here, we investigate the effect of the bucket size in terms of ROUGE scores (the arithmetic mean of R-1, R-2, and R-L) and inference efficiency in the unsupervised setting where training targets are 8-word summaries from [33], following Table 1.

As seen in Figure 2, the ROUGE score continuously decreases with a larger bucket size (thick orange curve). This not only confirms the inexactness of our algorithm, but also shows that a small bucket size does not hurt the performance much. On the other hand, the inference time decreases drastically at the beginning (thin blue curve) because we have fewer dynamic programming steps; as the bucket size increases, the inference time convergences to NACC without length control. Based on this analysis, we set the bucket size to be 4 in our experiments.

**Length-Transfer Generation.** Our NACC model is capable of length-transfer generation, that is, generating summaries of different lengths from the training targets. Such generation is important to real-world applications where summaries of various lengths are needed for the same input, e.g., fitting different screen widths. Although generating a short enough summary may satisfy all possible length constraints, a longer summary that better utilizes the length budget can preserve more information; this is also reflected by ROUGE scores, which prefer longer summaries, as argued by [33].

Figure 3 compares the performance of NACC with Su et al. [34] when learning from 8-word summaries in the unsupervised setting. When the inference length budget is less than training ($x$-axis $< 50$), the ROUGE score of NACC decreases almost linearly with a decreasing length budget, but Su et al. [34]'s approach degrades faster than ours. For $x$-axis $> 50$, we find the soft penalty in [34] is unable to generate longer summaries than trained (shown by the dashed orange line), whereas our approach is able to utilize the increased length budget and achieve higher ROUGE scores.

**Case Study.** Table 5 shows example summaries generated by NACC and the AR Transformer on the Gigaword test set.

As seen, a model without length control may generate a summary that happens to have the desired length (AR Transformer in the supervised setting), a shorter summary (NACC in the supervised setting), or a longer summary (both AR Transformer and NACC in the unsupervised setting). A shorter summary fails to utilize the entire length budget, leading to less information conveyed, whereas a longer summary requires truncating for explicit length control. Both cases are undesired.

By contrast, our proposed algorithm is able to generate a summary whose length is close to but less than the length budget. The resulting summary is more complete than truncating, and better keeps the key information.

# 5    Conclusion

In this work, we propose a Non-Autoregressive Summariation model with Character-level length Control (NACC), which can explicitly control the number of characters in the predicted summaries. Experiments show that our NACC approach not only achieves the state-of-the-art non-autoregressive (NAR) performance on Gigaword and DUC2004 datasets under length constraints, but is several times faster than autoregressive (AR) models and even outperforms an AR Transformer in the unsupervised

setting. Moreover, our approach is able to perform length-transfer generation, that is, generating summaries of different lengths from the training set.

**Limitation and Future Work.** Our proposed length-control algorithm only works with non-autoregressive models, which is a limitation (but not a disadvantage) of our work. We have clearly stated this throughout the paper.

Our NACC approach—although largely outperforming previous NAR models and retaining high inference efficiency—achieves lower ROUGE scores than AR models in the supervised setting. This is the limitation of all NAR approaches in general, and can be addressed by future work. We nevertheless outperform AR models in the unsupervised setting.

## Appendices

The full paper, including appendices, is available at `https://arxiv.org/abs/2205.14522`.

## Acknowledgments

We thank all reviewers for their valuable comments. The research is supported in part by the Natural Sciences and Engineering Research Council of Canada (NSERC) under grant No. RGPIN2020-04465, the Amii Fellow Program, the Canada CIFAR AI Chair Program, a UAHJIC project, a donation from DeepMind, and the Digital Research Alliance of Canada (alliancecan.ca).

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
