# A  Proof of Theorem 1

**Theorem 1.** *(1) If the bucket size $\alpha = 1$ and consecutive repetitions are not merged, then $\mathbf{d}^{S,T}$ is the most probable sentence of $T$ characters given by the $S$ prediction slots. (2) If $\alpha \neq 1$ or repeating tokens are merged, our algorithm may not be exact.*

*Proof.* [Part (1)] Our NACC is trained by the Connectionist Temporal Classification (CTC) algorithm [11], which merges repeated consecutive tokens and removes $\epsilon$s in the output sequence. Since the merging operation establishes dependencies between tokens in the output sequence, our length-control algorithm is inexact.

In this part, we consider a variant of the CTC algorithm that does not merge repeated tokens but only removes $\epsilon$s; we denote this modified reduction operation by $\Gamma'$. For example, $\Gamma'(aa\epsilon abb\epsilon) = aaabb$. Our thus revised algorithm works as follows.

We denote $\widetilde{\mathbf{d}}^{s,l} = \widetilde{\mathrm{d}}_1^{s,l} \cdots \widetilde{\mathrm{d}}_s^{s,l}$ as the recursion variable, being the most probable $s$-token sequence that is reduced to a summary of length $l$.

The initialization of $\widetilde{\mathbf{d}}^{s,l}$ is the same as the original length-control algorithm (§3.2), since the merging operation is not involved here. However, the recursion involves only two cases:

- Case 1: $\mathrm{w}_s = \epsilon$. The recursion of this case is also the same (see Eqn. 4):
$$\widetilde{\mathscr{D}}_1^{s,l} = \left\{ \widetilde{\mathbf{d}}^{s-1,l} \oplus \epsilon \right\} \tag{8}$$

- Case 2: $\mathrm{w}_s \neq \epsilon$. We have a set of candidate sequences:
$$\widetilde{\mathscr{D}}_2^{s,l} = \left\{ \widetilde{\mathbf{d}}^{s-1,l'} \oplus \mathrm{w}_s : \left( u(\mathrm{w}_s) + \sum_{\mathrm{d} \in \widetilde{\mathbf{d}}^{s-1,l'}} u(\mathrm{d}) \right) = l, \mathrm{w}_s \neq \epsilon, \text{ and } l' < l \right\} \tag{9}$$

  This is analogous to Eqn. (6), where $\alpha = 1$ (due to our theorem assumption). Also, the condition $\mathrm{w}_s \neq \widetilde{\mathrm{d}}_{s-1}^{s-1,l'}$ in Eqn. (6) is dropped here because this algorithm variant does not merge repeated tokens.

Then, the algorithm chooses the most probable candidate sequence as $\widetilde{\mathbf{d}}^{s,l}$, given by

$$\widetilde{\mathbf{d}}^{s,l} = \operatorname*{argmax}_{\mathbf{d} \in \widetilde{\mathscr{D}}_1^{s,l} \cup \widetilde{\mathscr{D}}_2^{s,l}} \sum_{s=1}^{S} v_s(\mathrm{d}_s) \tag{10}$$

Now we will prove that the algorithm is exact: suppose $P_{s,l} := \sum_{i=1}^{s} v_i(\widetilde{\mathrm{d}}_i^{s,l})$ is the log probability of $\widetilde{\mathbf{d}}^{s,l}$, we have

$$P_{s,l} = \max_{\mathrm{d}_1 \cdots \mathrm{d}_s : |\Gamma'(\mathrm{d}_1 \cdots \mathrm{d}_s)| = l} \sum_{i=1}^{s} v_i(\mathrm{d}_i) \tag{11}$$

In other words, $\widetilde{\mathbf{d}}^{s,l}$ is the most probable $s$-token sequence that is reduced to length $l$. This is proved by mathematical induction as follows.

**Base Cases.** For $l = 0$, the variable $\widetilde{\mathbf{d}}^{s,0}$ can only be $s$-many $\epsilon$s. The optimality in Eqn. (11) holds trivially.

For $s = 1$ but $l > 0$, the algorithm chooses $\widetilde{\mathrm{d}}^{1,l} = \operatorname*{argmax}_{\mathrm{d}_1 : u(\mathrm{d}_1) = l} v_1(\mathrm{d}_1)$. Therefore, $P_{1,l} = \max_{\mathrm{d}_1 : |\Gamma'(\mathrm{d}_1)| = l} v_1(\mathrm{d}_1)$, showing that Eqn. (11) is also satisfied with only one term in the summation.

**Induction Step.** The induction hypothesis assumes $P_{s-1,l'} = \max_{\mathrm{d}_1 \cdots \mathrm{d}_{s-1} : |\Gamma'(\mathrm{d}_1 \cdots \mathrm{d}_{s-1})| = l'} \sum_{i=1}^{s-1} v_i(\mathrm{d}_i)$ for every $l' < l$. We will show that the algorithm finds the sequence $\widetilde{\mathbf{d}}^{s,l}$ with $P_{s,l} = \max_{\mathrm{d}_1 \cdots \mathrm{d}_s : |\Gamma'(\mathrm{d}_1 \cdots \mathrm{d}_s)| = l} \sum_{i=1}^{s} v_i(\mathrm{d}_i)$.

| Word | $P_1(\cdot|\mathbf{x})$ | $P_2(\cdot|\mathbf{x})$ |
|------|------|------|
| I | 0.3 | 0.1 |
| am | 0.4 | 0.6 |
| a | 0.2 | 0.05 |
| $\epsilon$ | 0.1 | 0.25 |

Table 6: A counterexample showing that our algorithm may be inexact if $\alpha \neq 1$ or repeated tokens are merged. Here, we set the vocabulary to be three words plus a blank token $\epsilon$.

According to Eqn. (10), the variable $\widetilde{\mathbf{d}}^{s,l}$ is the most probable sequence in $\widetilde{\mathscr{D}}_1^{s,l} \cup \widetilde{\mathscr{D}}_2^{s,l}$. Thus, we have

$$P_{s,l} = \max_{l',\mathrm{d}_s:l'+u(\mathrm{d}_s)=l} \{P_{s-1,l'} + v_s(\mathrm{d_s})\} \tag{12}$$

$$= \max_{l'} \left\{ P_{s-1,l'} + \max_{\mathrm{d}_s:l'+u(\mathrm{d}_s)=l} v_s(\mathrm{d_s}) \right\} \tag{13}$$

$$= \max_{l'} \left\{ \max_{\mathrm{d}_1\cdots\mathrm{d}_{s-1}:|\Gamma'(\mathrm{d}_1\cdots\mathrm{d}_{s-1})|=l'} \sum_{i=1}^{s-1} v_i(\mathrm{d}_i) + \max_{\mathrm{d}_s:l'+u(\mathrm{d}_s)=l} v_s(\mathrm{d_s}) \right\} \tag{14}$$

$$= \max_{l'} \left\{ \max_{\substack{\mathrm{d}_1\cdots\mathrm{d}_s:\\ |\Gamma'(\mathrm{d}_1\cdots\mathrm{d}_{s-1})|=l'\\ |\Gamma'(\mathrm{d}_1\cdots\mathrm{d}_s)|=l}} \sum_{i=1}^{s} v_i(\mathrm{d}_i) \right\} \tag{15}$$

$$= \max_{\mathrm{d}_1\cdots\mathrm{d}_s:|\Gamma'(\mathrm{d}_1\cdots\mathrm{d}_s)|=l} \sum_{i=1}^{s} v_i(\mathrm{d}_i) \tag{16}$$

Here, (13) separates the $\max$ operation over $l'$ and $\mathrm{d}_s$; (14) is due to the induction hypothesis; (15) holds because the two $\max$ terms in (14) are independent given $l'$, and thus the summations can be grouped; and (16) further groups the two $\max$ operations with $l'$ eliminated. The last two lines are originally proved in [14] and also used in [7].

[Part (2)] We now prove our algorithm may be inexact if $\alpha \neq 1$ or repeated tokens are merged. We show these by counterexamples.[4]

Suppose $\alpha \neq 1$ and in particular we assume $\alpha = 2$. We further assume repeated tokens are not merged. Consider the example shown in Table 6. The length-control algorithm finds $\widetilde{\mathbf{d}}^{1,1} = \{\text{"am"}\}$, and then $\widetilde{\mathbf{d}}^{2,2} = \{\text{"am I"}\}$ with the probability of $0.4 \cdot 0.1 = 0.04$, as the first bucket covers the length range $[1, 2]$ and second $[3, 4]$. Here, we notice that two words are separated by a white space, which also counts as a character). However, the optimum should be $\{\text{"I am"}\}$, which has a probability of $0.3 \cdot 0.6 = 0.18$.

Now suppose repeated tokens are merged, and we further assume the length bucket $\alpha = 1$ in this counterexample. Again, this can be shown by Table 6: the algorithm finds $\mathbf{d}^{1,1} = \{\text{"I"}\}$ and $\mathbf{d}^{1,2} = \{\text{"am"}\}$, based on which we have $\mathbf{d}^{2,3} = \{\text{"I a"}\}$ with probability $0.3 \cdot 0.05 = 0.015$. However, the optimum should be $\{\text{"a I"}\}$ with probability $0.2 \cdot 0.1 = 0.02$.

$\square$

The above theoretical analysis helps us understand when our algorithm is exact (or inexact). Empirically, our approach works well as an approximate inference algorithm.