# OpenReview forum: "A Character-Level Length-Control Algorithm for Non-Autoregressive Sentence Summarization"
_NeurIPS.cc/2022/Conference — NeurIPS 2022 Accept_

### Official Review · Reviewer_5WDy · 2022-07-10

**Rating:** 6
**Confidence:** 4
**Soundness:** 3 good
**Presentation:** 3 good
**Contribution:** 2 fair

**Summary:**

This paper proposed a dynamic programming algorithm based on the Connectionist Temporal Classification (CTC) model for controlling the  summary length in text summarisation tasks. This is important in many tasks such as headline generation. Experiments on DUC2004 and Gigaword showed the effectiveness of this method. This paper might be more appropriate for NLP conferences.


**Questions:**

For unsupervised text summarisation, there is also a length control mechanism, e.g. do clustering for sentences and choose the centroid or do text compression for each cluster, so the number of clusters is proportional to the summary length. See [1]. Apart from the truncate methods, it would be better to add such baseline methods.


[1] Zhao, J., Liu, M., Gao, L., Jin, Y., Du, L., Zhao, H., Zhang, H. and Haffari, G., 2020, July. Summpip: Unsupervised multi-document summarization with sentence graph compression. In Proceedings of the 43rd international acm sigir conference on research and development in information retrieval (pp. 1949-1952).

**Limitations:**

* The domain of the datasets are limited to news.
* Some experiments with large pre-trained models (e.g. T5, BART) and CTC may be considered.

**Strengths And Weaknesses:**

Strengths:
* Leverage the CTC method for controlling the summary length in text summarization.

Weakness:
* For evaluation ROUGE is considered limited, some other semantic focused metrics like BertScore may also be considered.
* DUC-2004 and Gigaword are small news datasets, the summaries are more lying in the first few sentences, it would be better if the authors can try larger datasets from other domains, e.g. arXiv scientific paper title generation.

---

> ### Author Response · Authors · 2022-07-31
> **Response to Reviewer 5WDy**
>
> _“This paper might be more appropriate for NLP conferences.”_
>
> Thanks for the suggestion. We submitted to NeurIPS because our contribution includes designing a theoretically understood algorithm based on dynamic programming. While the application focus of this paper is summarization, we believe the idea may be generalized to different text generation tasks (e.g., generating longer and more informative dialogue responses) and other non-NLP domains (generating DNA sequences of a certain length). We’ll point to these future directions in the revision.
>
> Weakness 1: _For evaluation ROUGE is considered limited, some other semantic focused metrics like BertScore may also be considered._
>
> We mainly adopted the standard metrics (namely, certain ROUGE variants) in each dataset. This makes it possible to compare our approach with previously published state-of-the-art models. We acknowledge that ROUGE scores may not be perfect, and thus conducted human evaluation on selected model variants, showing consistent results with ROUGE scores.
>
> We understand that BERTScore is a more recent and powerful metric. However, we do have concerns when it is applied to our approach, as we also use an encoder-only Transformer, having the structure as BERT. Thus, it becomes both an athlete and a judge. We’ll use BERTScore when our model is of a different structure (e.g., GPT-style autoregressive models).
>
> Weakness 2: _DUC-2004 and Gigaword are small news datasets, the summaries are more lying in the first few sentences, it would be better if the authors can try larger datasets from other domains, e.g. arXiv scientific paper title generation._
>
> Thank you for the suggestion. Again, we chose Gigaword and DUC2004 in order to compare with previous state-of-the-art methods. Schumann et al. (2020) perform hundreds of local search steps for each sample, and their inference is very slow. We’re happy to try different datasets as future work (probably as a journal extension should the paper be accepted).
>
> Question 1:_ For unsupervised text summarisation, there is also a length control mechanism, e.g. do clustering for sentences and choose the centroid or do text compression for each cluster, so the number of clusters is proportional to the summary length. See [1]. Apart from the truncate methods, it would be better to add such baseline methods._
>
> Thank you for the suggestion. It appears that choosing the number of clusters does not control the number of characters explicitly, so truncating is still needed.
>
> Nevertheless, we did extend Schruman et al. (2020)’s word-level search approach, and performed character-constrained discrete search. The results are better than Schruman et al. (2020)’ word search, but much worse than our explicit length control.
>
>
> Limitation 1: _The domain of the datasets are limited to news._
>
> Thanks. See weakness 2.
>
> Limitation 2: _Some experiments with large pre-trained models (e.g. T5, BART) and CTC may be considered._
>
> Thanks for the suggestion. Our model is an encoder-only architecture, and thus we tried BERT as the pretrained model. However, it did not improve much. We hypothesize that this is because BERT is pretrained for representation learning, whereas our model (although having the same structure) is for non-autoregressive generation. We’re willing to research NAR pretraining techniques, but it goes beyond the scope of this paper.
>
> In summary, we thank the reviewer for the inspiring comments. We will discuss them in the revision, and continue this line of research as future work.

---

### Official Review · Reviewer_BRgS · 2022-07-11

**Rating:** 6
**Confidence:** 5
**Soundness:** 3 good
**Presentation:** 3 good
**Contribution:** 3 good

**Summary:**

This paper investigates the problem of length-controlled sentence summarization and proposes NACC, a Non-Autoregressive summarization model with Character-level length Control based on a knapsack-like algorithm. The authors conduct an evaluation of the Gigaword headline generation and DUC2004 datasets in both supervised and unsupervised settings and the experimental results show that the proposed framework can obtain comparable results to the baseline methods.


**Questions:**

na

**Limitations:**

The authors  addressed the limitations and potential negative societal impact of their work.

**Strengths And Weaknesses:**

Strength:
1. Length-controlled text summarization is a useful task in practice especially meets the headline generation tasks.
2. The motivation and assumption are reasonable. A knapsack-like problem setting and non-autoregressive framework are interesting. The proposed method is straightforward and easy to be implemented.
3. The experimental comparisons and analysis are sufficient and human evaluation is also conducted.

Weakness:
1. This work only conducts experiments on Gigawords and DUC2004, more experiments on more datasets are more concrete.
2. From the results tables we could observe that the results are much worse than some traditional auto-regressive methods. For example, on the test set of Gigawords, previous works can obtain some better results: 36, 17, and 33 for R-1, R-2, and R-L respevtively. Therefore I am wondering about the real quality of the generated sentences, though the length is a restricted condiction, the quality is more crucial.

---

> ### Author Response · Authors · 2022-07-31
> **Response to Reviewer BRgS**
>
> Thanks for highlighting the usefulness of our task and the interestingness of our proposed algorithm.
>
> Weakness 1: _This work only conducts experiments on Gigawords and DUC2004, more experiments on more datasets are more concrete._
>
> Thank you for the suggestion. Our paper addresses sentence summarization, because it is one of the key research problems in the entire summarization field. For example, summarization of a long document can be accomplished by first extracting salient sentences and then performing sentence summarization (Lebanoff et al, ACL2019. Xu et al, EMNLP2019. Mendes et al, NAACL2019 ).
> Gigawords and DUC2004 are commonly used in the setting of sentence summarization (Schumann et al., 2020 ); therefore, we also adopt them for comparison with state-of-the-art models. We’re eager to extend our approach to other settings as future work.
>
> Weakness 2: _From the results tables we could observe that the results are much worse than some traditional auto-regressive methods. For example, on the test set of Gigawords, previous works can obtain some better results: 36, 17, and 33 for R-1, R-2, and R-L, respectively. Therefore I am wondering about the real quality of the generated sentences, though the length is a restricted condition, the quality is more crucial._
>
> Our non-autoregressive model, in fact, outperforms autoregressive models in the unsupervised setting (where the pseudo-groundtruth are mostly extracted words). Admittedly, our performance is lower than AR in the supervised setting (where the ground-truth summaries are more flexible). However, the difference isn’t too large: around 1-3 points in ROUGE scores.
>
> As mentioned in the “Limitation” section, NAR models generally perform worse than AR ones in exchange for time efficiency. However, the advances in the past several years have largely improved NAR performance, and we expect the gap will be smaller in the near future. The focus of our paper is the length-control algorithm, which is “orthogonal” to NAR model architecture per se. We’re willing to try out future NAR models when developed.
>
> We’re grateful for the reviewer’s comments and will incorporate the suggestions in the revision.

---

### Official Review · Reviewer_dLJR · 2022-07-12

**Rating:** 7
**Confidence:** 4
**Soundness:** 4 excellent
**Presentation:** 3 good
**Contribution:** 3 good

**Summary:**

The authors proposed a method to control the summary length by characters in non-autoregressive sentence summarization models. The proposed method utilizes dynamic programming-based decoding to follow the length limitation as an imposed constraint. Since considering all possible states in the decoding is intractable, the authors separately consider the length states by relying on buckets. Experimental results showed that the proposed method could control the summary length by keeping ROUGE scores. Furthermore, human evaluation results also showed that the proposed method outperformed the truncation-based length controlling.

**Questions:**

- In experiments, you prepared four buckets to reduce computational time in decoding. However, the inference time of a single bucket shown in Figure 2 seems practical. How did you decide on the bucket size in the experiments?
- Comparison with the autoregressive models in the unsupervised setting showed that NACC achieved comparable ROUGE scores with the AR methods. It indicates that NACC is effective in terms of informativeness. On the other hand, the current setting lacks the comparison between the proposed and AR methods in terms of fluency. Is there any information about this point?

**Limitations:**

The proposed method NACC is on the basis of non autoregressive decoding, which may be inferior to autoregressive decoding in fluency.

**Strengths And Weaknesses:**

Strength:
- Exact length control based on dynamic programming.
- Length can be controlled while maintaining summary performance.

Weakness:
- Lack of exact solution due to the use of buckets for the proposed method to run in practical time.
- Lack of human evaluation for the proposed method and conventional AR methods.
- Controlling summary length by a non-autoregressive model has already been conducted in the deletion-based sentence compression task. [1]

[1] Wang, Liangguo, et al. "Can Syntax Help? Improving an LSTM-based Sentence Compression Model for New Domains." Proceedings of the 55th Annual Meeting of the Association for Computational Linguistics (Volume 1: Long Papers). 2017.

---

> ### Author Response · Authors · 2022-07-31
> **Response to Reviewer dLJR**
>
> We thank the reviewer for recognizing our contribution of proposing a dynamic programming algorithm for length control.
>
> Weakness 1: _Lack of exact solution due to the use of buckets for the proposed method to run in practical time._
>
> We analyzed the exactness and inexactness of our algorithm in Theorem 1 (proof in appendix). We acknowledge that the bucket size may make the algorithm inexact. However, the decrease in performance is imperceivable when the bucket size is small, but the computational overhead of DP also drops significantly. Based on Fig 2, we determine that a moderate bucket size (e.g., 4) achieves both high summarization quality and high inference efficiency.
>
> Weakness 2: _Lack of human evaluation for the proposed method and conventional AR methods._
>
> Thanks for the comment. Due to the time and resource constraints, we prioritized the studies on our proposed length-control algorithm. The main argument of this paper is not our NAR model outperforms AR models, and thus human evaluation on AR is not central to our conclusion. Nevertheless, our human evaluation shows consistent results with automatic metrics, which can give an overall comparison between AR and NAR models.
>
> Weakness 3: _Controlling summary length by a non-autoregressive model has already been conducted in the deletion-based sentence compression task._
>
> We thank the reviewer for the comment. Wang et al. (2017) address extractive summarization, because they delete certain words and thus the remaining words are extractive. We instead perform length control in the setting of generative summarization.
>
> Further, our approach is able to control the length at the character level. Extending word-level control to character-level control is nontrivial (because we have to look into a word), and this is precisely the contribution of our paper, as we model it as a knapsack-like problem.
> In our experiments, we extended a more recent paper (Schumann et al., 2020) and performed character-constrained discrete search (add and delete), which is more general than the recommended paper (delete only). Our performance is much higher than constrained search (Tab 1).
> We are grateful for the suggested literature and will include it for discussion in the revision.
>
> Questions 1: _In experiments, you prepared four buckets to reduce computational time in decoding. However, the inference time of a single bucket shown in Figure 2 seems practical. How did you decide on the bucket size in the experiments?_
>
> We select a bucket size of 4 because the inference time drops significantly before this size while the ROUGE scores don't vary much.
> Compared with the single bucket, a bucket size of 4 decreases the latency by at least 20% (from over 0.03 to 0.024) but does not change the ROUGE score much (slightly over 20 for both settings).
>
> Question 2 : _Comparison with the autoregressive models in the unsupervised setting showed that NACC achieved comparable ROUGE scores with the AR methods. It indicates that NACC is effective in terms of informativeness. On the other hand, the current setting lacks the comparison between the proposed and AR methods in terms of fluency. Is there any information about this point?_
>
> Thank you for the question.
>
> Admittedly, the non-autoregressive model generates (slightly) lower-quality text in exchange for large efficiency improvements. This is also true for our model, as our DP’s computationally overhead is negligible.
>
> In our experiments, we found that the disfluency of our NAR length-controlled output is usually local and minor, whereas truncating over-lengthed summaries (given by either AR or NAR models) will make the sentence incomplete and sometimes hard to understand, as shown by our case study.
>
> We thank the reviewer for the insightful comments. We’ll address them in the revision.

---

### Official Review · Reviewer_aypC · 2022-07-17

**Rating:** 4
**Confidence:** 4
**Soundness:** 3 good
**Presentation:** 3 good
**Contribution:** 1 poor

**Summary:**

The paper proposes a character-length-constrained sentence summarization model by extending the length constraint from word-level to character-level in inference. The major technical part follows NAUS [Liu et al, ACL’22] which is a BERT-encoder with a CTC decoder. The main contribution is extending the NAUS’s word length constraint from word level to character level, which is formalized as a standard 0-1 knapsack problem with a DP solver. The experiments on Gigaword headline generation and DUC2004 datasets demonstrate the performance over several truncation models in terms of ROUGE scores.

**Questions:**

Please see Limitations.

**Limitations:**

1. The bucket grouping trick (lines 142-147, page 4) has some limits: (1) it would not reduce the theoretical complexity, i.e., quadric remaining as the 2D programming; (2) it may suffer from heavy collisions, as the skewed long-trail distributed character counting makes most words share the same bucket.

2. The idea and the technical part seem quite similar to NAUS (both using DP solution), it would be better to clearly state what the main novelty is compared with word-level constraints (not only with 0-1 knapsack problem) and in what points make methodologically is challenging.

3. Besides the truncation model, the method should be compared with the non-truncation model, i.e., NAUS.

Minor issue:
In Theorem 1. “See the appendix for the proof.” -> no appendix found.

**Strengths And Weaknesses:**

Pros:
The write-up is good, and the paper is easy to follow

Cons:
1. The character-level constraint for summarization is not well-motivated. In common sense,  character counts are positively correlated with character counts, especially considering most words share similar character counts as the long-tail distribution.

2. Limited technical novelty. The model heavily follows NAUS’s ([Liu et al, ACL’22]) DP-based word-length-constraint method. The idea of extending from word-level to character-level is trivial.

3. Lacks comparison with the non-truncation model, i.e., NAUS.

---

> ### Author Response · Authors · 2022-07-31
> **Response to Reviewer aypC (Part 1)**
>
> We thank the reviewer for the comments.
>
> Con 1: The character-level constraint for summarization is not well-motivated.
> We do not agree with the reviewer. The character-level constraint is motivated from multiple perspectives:
> 1. Our setting has real application values. Think of displaying a news summary on a mobile app. It will be quite annoying if the summary intrudes a new line by only a few words/characters. Likewise, a commercial LED display requires the exact number of characters, or otherwise, the text will be scrolling, making it difficult to read. These, unfortunately, are likely to happen if we only do word-level control, despite the positive correlation between word and character lengths.
> 2. The setting has algorithmic/theoretical inspirations. Although our character-level length control is an extension of word-level length control, we developed a different algorithm with theoretical justifications. (See below for the details of our algorithmic/theoretical contribution.)
> 3. Pragmatically, character-level length control is a common evaluation setting for summarization systems. For example, the DUC evaluation requires the maximum target length to be 75 bytes (https://duc.nist.gov/duc2004/). Such an important setting, unfortunately, is not adequately addressed in previous summarization studies. Our work largely bridges the gap between the summarization methodological research and its evaluation.
> Overall, the usefulness of the character-level control setting is acknowledged by all other reviewers. We’ll better articulate the motivations in the revision.
>
> Con 2/Limitation 2: “Limited technical novelty”, “it would be better to clearly state what the main novelty”
> We acknowledge that our character-level length control is an extension of the previous work’s word-level length control [Liu et al., 2022]. We have heavily cited the previous work and honestly discussed their contribution throughout our paper.
> The main novelty of our paper include:
> 1. We address precise character-level length control for summarization, which we (and other reviewers) believe is a useful and interesting setting (see the response to Con 1).
> 2. Our character-level length-control algorithm is unique. Apparently, it differs from the word level as we have to “look into” a word. Therefore, we formulate and solve it as a knapsack-like problem.
> The reviewer then believes we’re simply applying “a standard 0-1 knapsack problem,” which is not true. A standard 0-1 knapsack problem has a fixed value function, whereas our value function evolves over time. Therefore, our dynamic programming is different from the “standard 0-1 knapsack problem” too. We have clearly stated this in our manuscript (after Theorem 1).
> 3. In this paper, we considered both supervised and unsupervised summarization, whereas Liu et al. (2022) are only about the unsupervised setting.
> Generally, our technical novelty involves algorithm design and analysis. We feel it’s a decent contribution in comparison with other published papers (e.g., predicting POS-tags for NAT summarization [Su et al., EACL 2021], encoding length information into positional encodings [Takase et al., NAACL19], and NAT pretraining [Qi et al., ICML21]).
>
> “what points make methodologically is challenging”
> Our algorithm is a unique design for character-level length control. The formulation of a knapsack-like problem (not the standard 0-1 knapsack) is novel, and maintaining inference efficiency is challenging. Thus, we proposed the bucket treatment which is not seen in Liu et al. (2022) either.

---

> > ### Author Response · Authors · 2022-07-31
> > **Response to Reviewer aypC (Part 2)**
> >
> > Con 3/Limitation 3: Lacks comparison with the non-truncation model, i.e., NAUS.
> > Thanks for the suggestion. NAUS (word-level control) still requires truncating, because the word level does not transfer to the character level directly. We evaluated NAUS (char-level truncating) as follows:
> >
> > |              |                        | ROUGE-1 | ROUGE-2 | ROUGE-L |
> > |:------------:|:----------------------:|:-------:|:-------:|:-------:|
> > |  Supervised  | NAUS (char-truncating) |   32.35 |   12.45 |   30.66 |
> > |              |    NACC (this paper)   |   33.66 |   13.73 |   31.79 |
> > | Unsupervised | NAUS (char-truncating) |   25.71 |    8.55 |   23.85 |
> > |              |    NACC (this paper)   |   27.45 |    8.87 |   25.14 |
> >
> > As seen, our approach is quite a few points higher than NAUS. This verifies the effectiveness of our approach, and confirms that our paper and Liu et al. (2022) are different approaches. We’ll include and discuss the results in the revision.
> >
> > Limitation 2: “bucket grouping trick (lines 142-147, page 4) has some limits”
> > (1) Although our bucket treatment does not reduce theoretical complexity, it largely reduces the overhead of our DP during inference (Fig 2), as it mostly converges to the base NAT model.
> > (2) We do not believe our DP algorithm suffers from heavy collisions. This can be understood both intuitively and empirically. First, our intuition tells us that, given the discreteness of the number of characters in a word, it is unlikely to generate a high-quality summary given the exact number of characters. Empirically, we find a bucket size up to 5 does not degrade the performance much (Fig 2). We further tried the beam search idea in Liu et al. (2022), but it does not help much in our work. All the evidence suggests that the bucket treatment is appropriate for character-level length control.
> >
> > “In Theorem 1. “See the appendix for the proof.” -> no appendix found”
> >
> > We submitted the appendix as a standalone pdf doc, which is explicitly allowed by the NeurIPS submission requirement. The link is here: https://openreview.net/attachment?id=KXybrIUJnya&name=supplementary_material
> >
> > In summary, our contributions include a new and interesting setting, a novel and theoretically understood algorithm, and good empirical performance, which are generally recognized by other reviewers.
> >
> > The reviewer gave an extradintory score of “3: Reject: For instance, a paper with technical flaws, weak evaluation, inadequate reproducibility and incompletely addressed ethical considerations.” However, we find none of these actually applies to our paper. We sincerely hope the reviewer could read our (long) response, check the neglected file, revisit our paper, and render a more convincing recommendation. Thank you.

---

> > > ### Comment · Reviewer_aypC · 2022-08-08
> > > **Rebuttal Update**
> > >
> > > Thanks for the efforts made in the rebuttal. I have carefully read the response. They address a large part of my concerns and I do not have further questions. I really appreciate your honest discussion with NAUS and the convincing comparison with NAUS.
> > >
> > > Regarding my initial comments, I wish to clarify that the initial rating of 3 is merely based on the "conceptual novelty", which, in my opinion, is below the standard to go in NIPS, NOT the reasons shown in the system.
> > >
> > > After reading your response and the comments of other reviewers, I agree that the problem is interesting and the method does have some improvement over NAUS. However, I still do not think the methodical contributions are "significant" enough, especially considering the proposed one show no stunning compared with NAUS, in both methodology and empirical results.

---

### Author Response · Authors · 2022-08-07
**Response to Reviewers**

Dear reviewers:

We thank your efforts in reviewing our paper and providing insightful suggestions. We have addressed all the concerns raised and will incorporate them in the revision.

With the reviewer-author discussion deadline approaching, we would highly appreciate it if reviewers could take a look at our responses. Should there be any further questions, please do not hesitate to ask us.

Thank you very much,

-Authors

---

### Meta-Review · Area_Chair_B5jw · 2022-08-25

**Recommendation:** Accept
**Confidence:** Certain

**Metareview:**

This paper addresses sentence compression by controlling the output by the number of characters. The proposal relies
on a non-autoregressive generation model interfaced with a dynamic programming algorithm where lengths are divided into buckets
for efficient inference. The proposed algorithm has advantages over previous methods, both computationally and in terms
of results. The authors have addressed the reviewers comments, and honestly discussed shortcomings and limitations.
It is also worth pointing out that they even have conducted a human evaluation study to assess the completeness of the summaries
produced by their length-controlled model. I would urge the authors to evaluate additional aspects of the output such as content, and
faithfulness to the input (i.e., are there hallucinations). Also the comparison should include an AR system and the gold upper bound.

**Award:**

No

---

### Decision · Program_Chairs · 2022-09-14

Accept